# How Does Supervision Technique Affect Research? Towards Sustainable Performance: Publications and Students from Pure and Social Sciences

Iszan Hana Kaharudin [1,2,*], Mohammad Syuhaimi Ab-Rahman [2,3], Roslan Abd-Shukor [4], Azamin Zaharim [3], Mohd Jailani Mohd Nor [5], Ahmad Kamal Ariffin Mohd Ihsan [3], Shahrom Md Zain [2,3], Afiq Hipni [2,3], Kamisah Osman [6] and Ruszymah Idrus [7]

1   Centre for Liberal Studies (CITRA), Universiti Kebangsaan Malaysia, Bangi 43600, Malaysia
2   Research Advancement and Strategic Planning (RASP), Universiti Kebangsaan Malaysia, Bangi 43600, Malaysia; syuhaimi@ukm.edu.my (M.S.A.-R.); smz@ukm.edu.my (S.M.Z.); afiqhipni@yahoo.com (A.H.)
3   Faculty of Engineering and Built Environment, Universiti Kebangsaan Malaysia, Bangi 43600, Malaysia; azami@ukm.edu.my (A.Z.); kamal3@ukm.edu.my (A.K.A.M.I.)
4   Department of Applied Physics, Universiti Kebangsaan Malaysia, Bangi 43600, Malaysia; ras@ukm.edu.my
5   Fakulti Kejuruteraan Mekanikal, Universiti Teknikal Malaysia (UTeM), Melaka 76100, Malaysia; jai@utem.edu.my
6   Department Teaching and Learning Innovation, Universiti Kebangsaan Malaysia, Bangi 43600, Malaysia; kamisah@ukm.edu.my
7   Department of Fisiology, Universiti Kebangsaan Malaysia, Bangi 43600, Malaysia; ruszy@medic.ukm.my
*   Correspondence: iszanhana@ukm.edu.my

**Abstract:** Supervision without effective monitoring and strategy planning can lead to zero output. The fear of productivity losses, combined with the horror of massively declining performance, has encouraged many leaders to increase their subordinates' monitoring efforts. This article explores the techniques used by lecturers in managing their research and how they affect the outcome and performance. Some techniques might sound familiar while some will be new. Two sets of questionnaires were developed to investigate the relationship between techniques and performance. One set was distributed to 15 lecturers and the other set was distributed to 100 students from different fields to get input from them on the best supervision characteristics. Fourteen outputs were outlined to show the weight of techniques used by the lecturers. From the survey results, lecturers who applied more than three techniques were able to produce more than three graduating students and more than 20 publications. The aim of the study is to list the most effective techniques for sustainable supervision which lead to output production.

**Keywords:** supervision skill; publication; research method; student output; sustainable monitoring technique

## 1. Introduction

Supervisors are an important asset to any business, organization, or institution. Supervisory skills are essential for every employee since they are the foundation for moving up in your career to leadership positions. Sustainable supervision describes supportive supervision as an approach to supervision that emphasizes joint problem-solving, mentoring, and two-way communication between supervisors and those being supervised towards producing impactful and continual outcomes. Active supervision is a proactive, low-intensity strategy to minimize challenging behaviors and increase desired behaviors [1]. Zappala et al. [2] mentioned supervisor support in their study as an expression of the support and interest that organizations have towards their subordinates, and the supervisor support may contribute towards enhancing the relation that subordinates have with their

organization and with the organizational changes that are introduced in it [3]. This is the factor of success of any organization. Therefore, developing sustainable supervisory skills can be a critical steppingstone for management and leadership positions which may lead to huge output production.

The recent urgency of producing articles worthy of high impact journals is leading to a new competition among lecturers and researchers worldwide. The number of publications does not reflect the greatness of an academic. It is the quality of a particular article that is sought [4]. This phenomenon does not only cover a certain field, for example engineering, but also other fields ranging from psychology and history to chemistry and mathematics. It is crucial for researchers to publish scientific articles because it is a key measurement of their performance, and will also enable them to get research grants, promotion, professional networking, and salary increment [5,6].

The multidisciplinary views illustrate, on one hand, the scientific affluence of this topic, but it raises, on the other hand, research limitations as it is difficult to reach a consensual concept of quality students and publications, as well as the underlying dimensions that should be used for its assessment. The most important thing is that publication in quality research journals is generally acknowledged to be a fundamental criterion of any research evaluation [7]. A perfect combination to evaluate the level of an academic paper are both citation indices and peer review, which have proven their worth as practicable methods of substituting research quality. However, Jones stated another method introduced by Ormerod [8–10] to evaluate the quality of research which is based on the analysis of content and derived from Reisman and Kirschnick [11]. Apart from publishing, there is another method that is considered crucial for academic development in the laboratory. Lecturers need to have a systematic approach to manage the people who are working in their research group [12]. As for the rest of the group, it is important that several methods are evaluated and tested first before being launched and practiced in order to determine their effectiveness. Basically, there are five key areas of research methods according to Burton and Schofield [13], which are (1) using research literature; (2) designing research studies; (3) analyzing data; (4) interpreting research findings; and (5) presenting the results.

Evaluation of an academic paper can be obtained by a number of methods and two of the most broadly used methods are citation indices and peer review [6,14]. A lecturer needs to understand both of these methods if they wish to have their articles ranked highly. Citation indices rely on bibliometric methods so there are several ways to increase the count such as co-operational citing, networking, and collaborating with more experienced and renowned researchers, and lecturers may urge the faculty to set up a system that will enable them to cross-reference other works by lecturers in the same institute or university. By confirming that the article has already possessed all the required criteria, this will ensure the article becomes "real" because then it would be available to the readers after being published [15]. This concept, however, does not appear to have truly taken root in the practice of higher education. To date, the literature continues to be "scant" where "measuring students' social and emotional predispositions before conducting research has not been widely studied, especially from a developmental standpoint" [16]. Studies describing first year undergraduate students' attitudes towards the sources of referenced articles in general and secondary research in particular must be resumed as a shift in generation approaches. This complication may not entirely conform to previous findings [17]. The following study, therefore, adds to the well-established macro research thread examining the relationship between information literacy and the affective domain and the less-established micro research thread describing student "social and emotional predispositions" towards the research laboratory. To this end, this study focuses specifically on general freshmen value attitudes towards the secondary research process. A comprehensive study needs to be performed to justify the reaction towards the results. As for the previous study, some results show that the success came from the notion where students and supervisors work together to achieve greatness in the research. Mutual dependence still exists no matter what is done to encourage the cooperation between students and supervisors [18]. This agreement must be

set up within the early stages of research to avoid any misunderstanding that could lead to disastrous results. Many cases reported that such cooperation would always lead to great achievements among the laboratory members. The problem is, however, closely related to the efficacy of the partnership. The partnership needs to be thoroughly understood by both parties to optimize the production of the laboratory. Furthermore, according to Meng et al. [19], three compulsory factors that act as main inputs for research activities are the number of research staff, the total research expenditures, and the research equipment expenditures. The need for this should be placed on a high level because the outcome of the research depends greatly on it. This is where the framework of ethics becomes beneficial to classify the importance of the research individually for the supervisor or as a whole, together with the rest of the laboratory members [20].

There are several sustainable techniques practiced by lecturers to monitor their research progress. Monitoring research can be defined as observing and inspecting the progress or quality of their research over a period of time [21]. This includes monitoring people involved with the research such as the laboratory assistants, students, and other collaborating researchers or lecturers. Some types of research are infinite and current where the methods suffer a major limitation related to data unavailability for several variables in the laboratory. This requires even more attention by the lecturers to ensure that the research is going smoothly. However, it has been a debate on how strictly the supervisors should monitor their students. Just like at the workplace, a research laboratory also feels to students like a place to perform all their experimentation, writing, editing, and so forth. It was reported by Oz et al. [22], that most employees in the workplace feel that constant monitoring can cause tension in the work environment. This can be applied in the research laboratory situation, so supervisors need to take a hard look at this fact because the performance depends greatly on the students. The performance is defined by including more substantial information to ensure that the relevance of subjects is obvious. Some suggested that the supervising system has a major impact on getting a more desired outcome from the experiment. This is not clear in producing as much impact as planned because there are other variables that must be taken into consideration. The more effort is put into the research, the more extra points researchers will be able to get on that. Mixed methods have also been popular with researchers who are willing to go to certain lengths to complete the research by combining qualitative and quantitative methods [23]. It is advised that a dedicated timeline is created to ensure that the research is smoothly executed. Some effective supervising methods are presentation, weekly progress report, and utilizing results as assessments. The presentation method will not only test students in their public-speaking skill but will also urge them to always be prepared and updated since the presentation is done weekly. The same goes for the weekly progress report, where students must submit weekly activities and progress on their research [3]. Supervisors need to come up with a standard format so that important details are not left out when the reports are submitted. Some results will show the level of success of the research and this indication is important in determining what steps are to be taken next. However, the consequences from these actions are obsolete, which indicate the severity of the issue at hand. More publications can be produced today. However, the number of high-quality ones is relatively low.

Apart from teaching, lecturers are also defined by their performance in publications. In order to produce publications, research needs to be done. However, an excellent, well-developed, and well-executed piece of research does not guarantee the equal level of quality publications. This is where writing and presenting skills in a scientific way are useful. Many researchers have issued warnings about using various techniques but were especially wary of qualitative approaches. Some opinions suggested that survey questionnaires, which is regarded as the most popular instrument for collecting consumer data, were susceptible to bias when used cross-culturally. However, they cautioned more strongly against the use of other techniques such as focus groups and consumer panels. He concluded that "it is difficult to justify the use of these instruments [focus groups and consumer panels] in all but the more developed Western societies. In countries where

the application of these instruments is somewhat dubious, market researchers would be better off using more direct observational approaches rather than being misled by gross errors of inference" [24]. It is noted that this area is of direct importance to international researchers and business executives, thus research has been conducted with those who have the everyday responsibility for getting results through international market research. These research practitioners are the people who can identify where qualitative techniques are used and the problems encountered in employing these techniques. Some studies suggested that this performance is likely to develop a real cause of bad research performance, which will affect other dimensions that are critical in determining the level of ranking. However, there are a few types of research that require a certain amount of commitment to really show their success, which focus on these individuals. The number of students undertaking Master's degrees is increasing, especially Master of Science degrees which have become immensely popular. Master's theses are required for the completion of these degrees which indicates more members of the academic staff are needed to perform supervisory tasks, including those with limited supervisory experience. Better Master's thesis results will determine the quality of academic achievement and future career prospects for these students, which will also reflect the supervisors' capability in handling their students and all the resources. Academic reputation is also at stake in the Master's thesis supervision because supervision process is included in evaluations during academic visitation and accreditation procedures.

In addition to the productivity of good quality publications, the number of graduated students with a first-class degree is equally important to lecturers' performance. Each academic level usually involves an individual final year project which emphasizes a specific area of knowledge or expertise. This project tests the student's ability to work on a relatively huge project on their own, with the assistance of one or two supervisors. Different supervisors have different supervising techniques, so students need to adapt to their supervisor's methods to make it work. There is a vast population of lecturers and researchers who are very good at designing and executing research and the findings are equally excellent. However, the most difficult part for them is to publish those findings in a good quality publication, especially in a top-rated journal [18]. This is why lecturers have to take note of several important aspects that can contribute to the success of their research findings being published. According to a BJP study in 1996, the journal ranking method by using citation analysis is not biased. It depends on the nation that publishes the articles [5]. The ways academics are networked into the academic circle are subjective and the situation is quite similar in every country. This makes it harder for authors from other countries, especially the developing ones, to publish in top journals as often as their peers in more developed countries. Students often find it difficult to communicate their research needs, which is where lecturers and supervisors play a very important role to emphasize critical thinking and strategizing on topic expansion [25]. This will encourage students to go out of their comfort zone and adapt to the research environment faster. The objective of this work is to determine the technique used by the researchers to manage their research. The questionnaire was developed to investigate the relationship between techniques and performance and distributed to 15 lecturers and 100 students from different fields to get input from them on the best supervision characteristics. Sharing the experiences on managing research/student projects would not only help in making the module more attractive and beneficial to the students but also in the consolidation of the guidelines and indicators of the proper supervision and a fair evaluation of this significant undergraduate study endeavor [26].

## 2. Concept of Research Performance

It is crucial to fully understand the concept of publications and students of quality. A publication that boasts quality is one that is packed with understandable and functional information, free of grammatical errors, and has the potential to boast a significant number of citations. The performance of student progress can be measured via number and quality of publications. It is reflective of the content and knowledge, idea presentation, deliberation,

critical thinking, etc. Most of the high-quality publications are reviewed by field experts. The excellence of an international publication is mostly based on peer review, peers being the ones inside the country and also overseas [14]. Section 3 will show the profile of high-quality research performance which will make it easier to understand the criteria in evaluating lecturers' performance. Methodology will be discussed in Section 4, and followed by the results and discussions in Section 5.

## 3. Profile of High-Quality Research Performance

Table 1 below shows the output for high-quality research management profile. These indicators (Symboled by 'O') will be defined the performance of the respondent based on the applied monitoring technique onto their research group. It is categorized into four (4) criteria such as publication, students, awards and funding. These 4 criteria determines the sustainability and visibility of the research group.

**Table 1.** Outputs for high-quality research management profile.

| | |
|---|---|
| Publication | (O1) Number of Q1 articles<br>(O2) Number of articles published in impacted journals<br>(O3) Number of articles published in indexed journals<br>(O4) Number of proceedings papers published<br>(O5) Number of chapters in books published<br>(O6) Number of books published |
| Students | (O7) Number of first-class undergraduates produced<br>(O8) Number of PhD graduates produced<br>(O9) Number of Master's graduates produced<br>(O10) Number of awards received from student's project |
| Awards | (O11) Number of international awards won<br>(O12) Number of national awards won<br>(O13) Number of faculty/department awards won |
| Funding | (O14) Average funding money spent (%) |

## 4. Methodology

There are two aims of this paper; the first is to evaluate the quality and quantity of publications produced by applying some defined techniques and the second is the quality and quantity of graduated students. Two different sets of questionnaires were designed to obtain these data and the participants were chosen from several faculties in a local research university. The first set of questions was distributed to 15 lecturers who were selected based on their performance. The other set was distributed to 100 students from numerous departments and faculties. The reason was to obtain pre-assumed different feedback from these students in order to know the differences in their needs. Feedback from these surveys is important because valuable data regarding the faculties' needs and accomplishment can be recorded and analyzed. Data were analyzed using spreadsheet software Microsoft Excel by using bar graphs to ensure convenience in viewing the results.

### 4.1. Survey for Students

The questionnaire for the students focused on their behaviors and views on the best research management techniques. Students from two different categories were chosen, pure science and social science. In this questionnaire, pure science encompassed of students from the science and technology, and engineering faculties. Social science encompassed of students from social science and humanities, economics and business, and language faculties. Each category received 50 sets of questionnaires. This division ensured that there would be sufficient comparison of data and this is one way to classify individual elements which are based on the pre-specified formats [27]. This is very crucial in obtaining knowledge from both streams because if only one stream were given the questionnaire, the result would be biased.

*4.2. Survey for Lecturers*

All the samples were selected from the faculty but different departments. Several limitations were encountered in obtaining data with this method, including confidentiality issue and the availability of samples. Only experienced and successful lecturers were chosen to ensure that the data would be just and significant. In the questionnaire, some research management techniques were listed. Some techniques might have sounded familiar and some were relatively new. This was to know what types of techniques have been applied by the lecturers in order to succeed in their respective fields. The questionnaire also instructed the lecturers to provide the numbers of publications and students produced by them. This shows the linkage between the methods and the performance achieved by lecturers which are the number of students graduated under them and the number of publications produced.

*4.3. Research Management Techniques Applied by Lecturers*

In the research laboratory, most of the studies are executed by the students who are supervised by the lecturer who acts as the head researcher. This means that the success of a particular study depends heavily on the students' work and effort. These social dynamics have a big impact on the research performance level [28]. Lecturers have a lot of commitments that require them to be out of the laboratory most of the time. Therefore, it is important to inculcate the sense of belonging in the students because it is the central key to the development of research and studies in a university [29]. In this paper, some new techniques are introduced, as well as other familiar methods to show that these techniques, which are devised by lecturers together with their students, help the research team to expand and succeed. The new research management methods have been applied by some lecturers and the production of students and publications have been recorded through the survey's result. Some techniques that will be explained later might be confused with the term methods. However, for the sake of simplicity, both terms will be defined as techniques.

- Technique 1: Weekly Presentations

This method requires the students to prepare a simple presentation regarding the progress on their research status. Several important aspects of the research must be included in the presentation to make it easier for the supervisor to evaluate the research development. Weekly presentations could help the supervisor to monitor whether the research is succeeding on the right track. This will prevent the students from executing the wrong research and could save time and money. The progress presentation can be done virtually or physically.

- Technique 2: Mentor–Mentee Program

Mentoring is a ubiquitous phenomenon in academia. Mentors in academia may act as teachers, sponsors, and/or collaborators [12]. Students are divided into several groups and a fellow lecturer will be appointed to head each group and act as their mentor in all relevant aspects. Even though some students will have different topics of research (but under the same faculty), this does not prevent the supervisor from advising them on other important tips. Students from social science and art backgrounds are more likely to identify other faculty mentors other than their supervisors compared to students from pure science backgrounds. The reason for this was mainly due to the research done by pure science students being mostly done in the laboratories which involve the handling of dangerous apparatuses and substances that require attentive guidance from specific professors.

- Technique 3: Logbook

A formatted logbook (either prepared by the supervisor at the beginning or prepared by the student and later approved by the supervisor) will be filled in by the students for every required detail regarding their project. The reason for the supervisor's approval regarding the format of the logbook is because crucial data on research progress need to be included or this technique will not be as effective as intended.

- Technique 4: Group Session

A group of students who have similar research topics and under the same discipline/field are supervised by the same supervisor and every meeting will involve all the students. This will encourage healthy debate and exchanging of points of views between the students, monitored by the supervisor. This will also lead to research integration with potentially new exploration and impactful findings.

- Technique 5: Freedom of Research

Students are on their own in executing the research in terms of time allocation, site determination, time arrangement, data analyzing, and so on. This does not mean that the supervisor will be completely ignoring the students. Minimal to regular meetings will be conducted to allow the supervisor to monitor the progress and whether the research is still following the objective.

- Technique 6: Culture of Excellence

Student motivation is widely recognized as a positive influence on engagement and learning in higher education [30]. There are four important elements in this technique which are monitoring, nurturing, motivation, and facilities. Monitoring includes the transformation process from progress report to technical paper and from proceeding to journal. Nurturing encompasses training and involving students in all activities such as book writing, laboratory management, and so on. Motivation focuses more on the role of lecturers as peers, mentors, counselors, and superiors to their students. Facilities include accessibility of computer laboratories, research grants, experimental tools, reference articles, and sources.

- Technique 7: Multi-Dimensional Assessment (MDA)

This approach is to maximize research results by viewing from as many points of views as possible. Having said that, different points of views require different kinds of people, which means a number of academics from different fields can get together and produce a number of publications from one research result [31,32]. Among the measures taken under this technique are listing the areas related to the research and obtaining views from experts in their respective fields. Figure 1 shows an example of different point of views derived from a single research result.

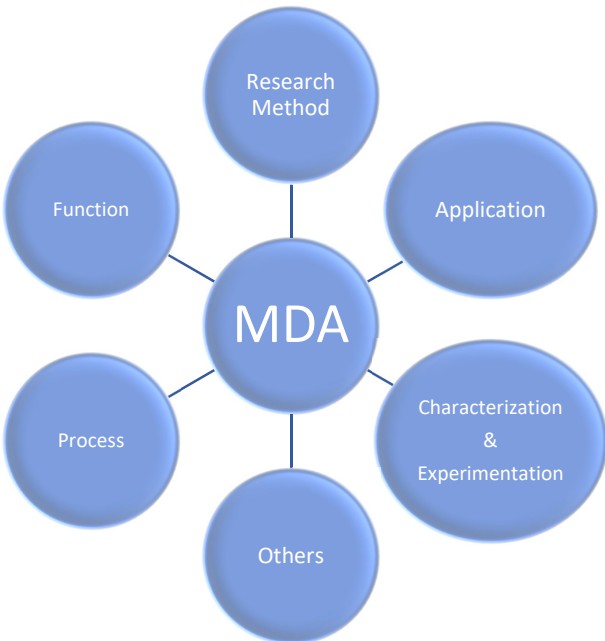

**Figure 1.** Aspects that can be derived from a single research work by implementing MDA.

Figure 2 shows possible new mini-topics that could be developed from research and user-access network security system. Every mini-topic might require academics with different expertise and, from the new starting point, other 'mini studies' could be resumed with other points of views. Table 2 shows the information that can be extracted from some of the mini-topics shown in Figure 2. Figure 3 shows the execution of MDA can be viewed from research topics and result analysis.

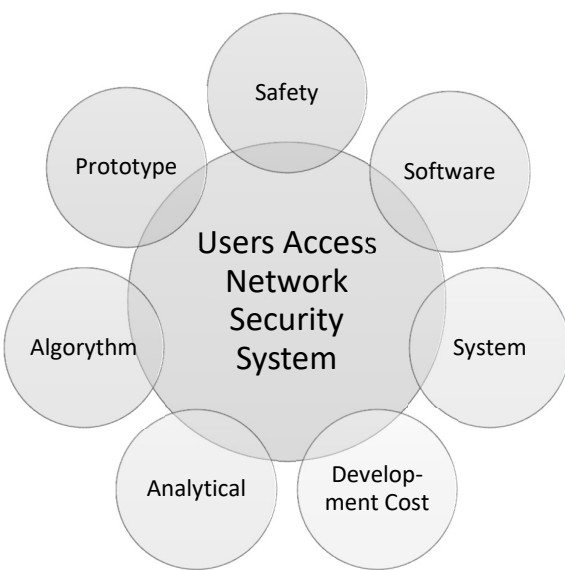

**Figure 2.** Important criteria derived from users.

**Table 2.** Explanation for mini topics obtained from Figure 1.

| Mini Topic (Point of View) | Subject |
| --- | --- |
| Software | Design and detail of the security system as a whole. |
| Device | Design and detail of the device or equipment that are being used to achieve the goal. |
| Development Cost | Cost to develop the system. |
| Failure/Damage Causes | Factors that could contribute to damages and failures in the system and ways to prevent them. |
| Security/Safety | Ability to generate security/safety issue for the network apart from working on the prototype design and suggestion mechanism. |
| System | Devices' interface process with the existing equipment. |
| Software | Programs that are developed to identify damage control. |

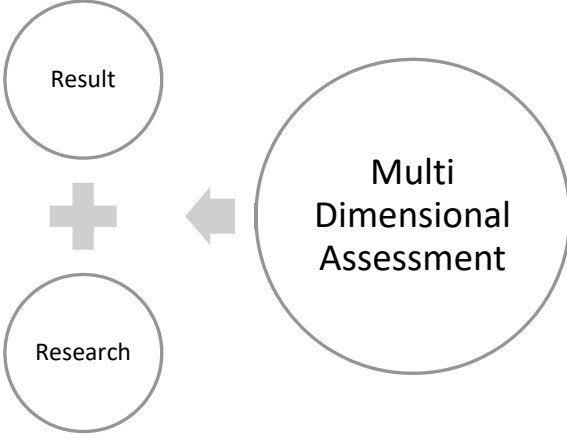

**Figure 3.** Execution of MDAs access network security system.

- Technique 8: Catalyst Activity

This technique includes organizing workshops and seminars that focus on devising strategies to maximize and enhance the quality of articles (a variety of views under the same research). Some activities that can be included under this technique are as follows:

i.     Publication catalyst workshop

This workshop, which is preferably organized outside of faculty, focuses on expanding one idea into several smaller ones and generating at as many publications as possible from the smaller ideas. This workshop usually takes three days. The first two days will mainly focus on completing the manuscript and, on the last day, some supervisors will read through the manuscripts and submit them to prospective journals.

ii.    Manuscripts refinement workshop

This workshop usually starts with a small talk regarding strategy or tips on good article writing followed by the refinement process of selected manuscripts submitted by participants. A few experienced and successful academics act as the inspectors, and everyone reads through the manuscript line by line. The inspectors point out some of the things that can be improved on and, at the end of the workshop, all selected publications are ready for submission.

iii.   High impact writing workshop

An almost complete manuscript is the 'ticket' for students to enter this workshop. This event is usually done in a casual way where the venue is probably a vacation spot and the activities throughout the program are divided into two, work and play. During work time, students are required to refine and complete their manuscripts by utilizing peer review, express presentation, and consulting with their supervisor. Play time is included to reduce the tension from completing the manuscripts. At the end of the workshop, usually on the third day, all the completed articles may be submitted to high-impact journals.

iv.    Group KPI (Key Performance Indicators) workshop

In every working community, KPI is crucial in evaluating productivity and quality of the work produced. However, most employees do not have a firm understanding of this value, which is likely to make this indicator negligible to them. This obviously does not help much with the productivity and quality factors. Therefore, this workshop serves two purposes; the first is to educate the employees on how this indicator works and the second is to improve the KPI for each employee by conducting several work-related activities during the workshop. In the academic environment, the most usual activity included is a publication-induced writing session. Publications include chapters in books, proceedings papers, articles for journals, and other related materials.

## 5. Results and Discussion

Based on the survey's result, it was obvious that students from different fields have different needs. This is considered important to determine other relevant factors such as fund allocation, research space requirement, research time period range, number of co-supervisors, and so on. Some data taken are based on the final statement given the range of borderline value that is synchronized. Some background studies reveal the functionality of the regional classification.

### 5.1. Results for Surveys on Students

Figures 4 and 5 show the results of factors that contribute to the quality of students' research. Facilities and financial factors score relatively higher for students from a pure science background as shown in Figure 4. This situation is due to the nature of research in this field where expensive machines and equipment need to be purchased and maintained, as well as the usage of exhaustive materials. This is not an indication that the students from a social science background do not need money to function effectively. Some courses

do need high financial support for expenses such as site trips, organized talks, seminars, and so on. Personal factor also scores slightly higher for students from a pure science background due to the relaxed nature of most studies in a social science background. Most of the research with a pure science background is scientific and experiment-based and the schedules are usually hectic. If personal problems really affect the quality of research, a mutual effort should be taken by both parties to resolve the situation. Other factors, including age, nationality, and gender have little effect on the quality of students' research. However, Figure 4 shows a higher score in these three factors compared to Figure 5. This is because these factors present the perception of the students of their supervisors.

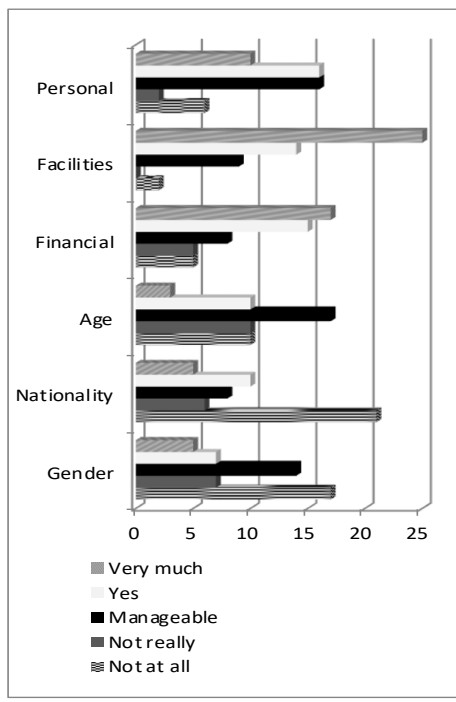

**Figure 4.** Factors that affect the quality of research of students from a pure science background.

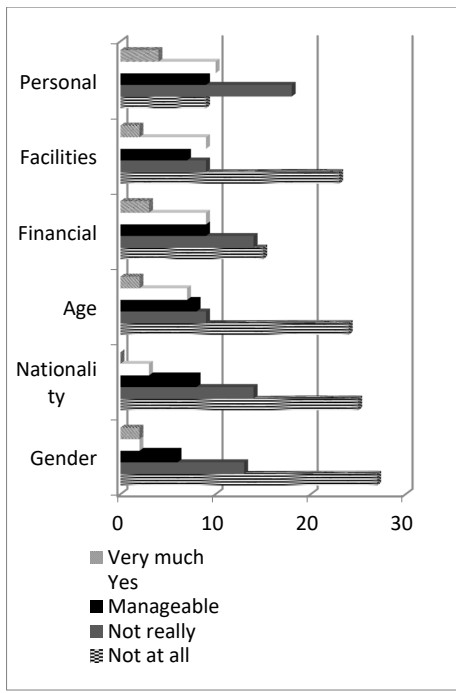

**Figure 5.** Factors that affect the quality of research of students from a social science background.

Figures 6 and 7 show the characteristics of a supervisor that can contribute to the success of their research, and a huge difference can be observed. Students from pure science need a supervisor who has all the characteristics listed in the questionnaire except one, which is very strict. Most of them answered highly ('Very much' and 'Yes') for these characteristics. Research that is scientific experimentation-based has a lot of variables to be taken into account and the results might be negative a lot of times before finally getting a slightly positive result. This is one reason why pure science students need supervisors who possess the agreed characteristics. The usage of complicated machinery and equipment requires the students to get assistance from either a supervisor or a laboratory technician. Furthermore, in the middle of the research period, some objectives of the research might need to be 'tweaked' a little to accommodate the current situation. For example, some tests always come back negative no matter the improvement. Perhaps the problem occurred due to the equipment's fault or weather unsuitability. This is why the characteristic 'Negotiable' also scored more highly for pure science students. For social science students, most answers obtained from them were in the 'Manageable' and 'Yes' categories. The results of their research are usually open-ended and subjective compared to their peers from pure science. This requires the students to be highly independent and able to think critically. Most research in this area requires little supervision from supervisors, which explains the lower score for characteristics in supervisors. Even though the characteristic 'Very strict' scored low for 'Very much', a high number of students from both and pure and social sciences still answered 'Yes' and the reason was because a strict supervisor could instill some 'fear' in the students and therefore, they are pushed to perform well in their research. Supervisors also need to have some openness according to Figures 6 and 7, because this would also help the students with their research. Openness is linked with friendliness and a friendly supervisor would be able to be closer with their students, almost in a peer-like relationship, which would enable them to discuss freely and have healthy arguments on their research.

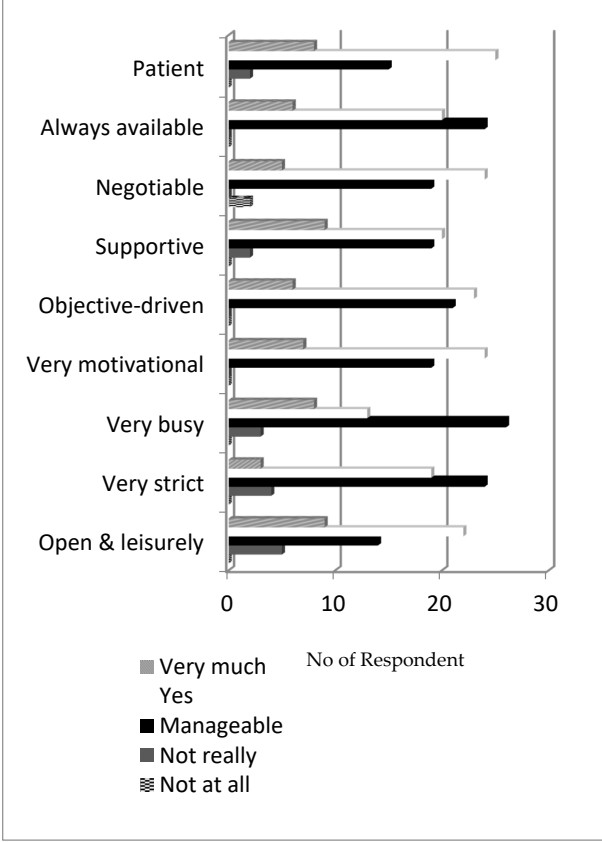

**Figure 6.** Characteristics in a supervisor that will drive students from a pure science background to produce a good research result.

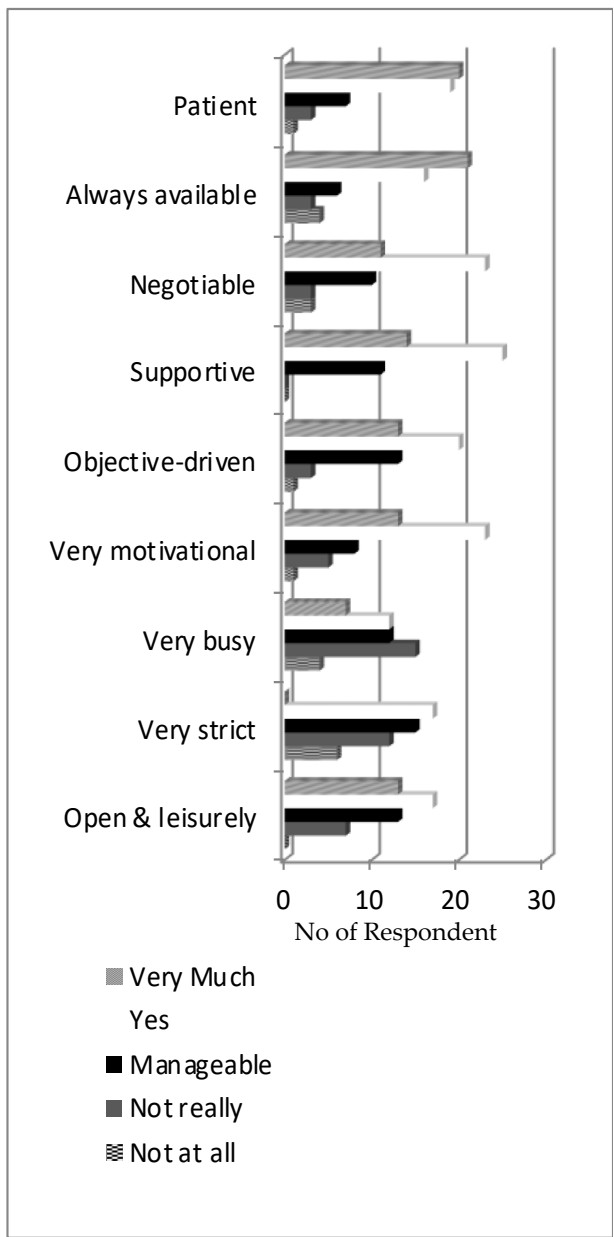

**Figure 7.** Characteristics in a supervisor that will drive students from a social science background to produce a good research result.

*5.2. Results for Surveys on Lecturers*

Figure 8 shows that the most common method used by excellent lecturers to supervise and monitor their students is face-to-face. This method is obviously the easiest way especially for students who are doing their research on campus. Lecturers can request a meeting to check the research progress and students can make an appointment for any enquiry or misunderstanding. The other three methods which are freedom of research, group meeting, and weekly presentation scored the second highest on average. Freedom of research usually applies to students who are also experts in their own respective fields and require little supervision. This does not mean that the supervisor totally ignores the research and lets the students go one hundred percent free. A group meeting is usually held when there is a mutual result required by the same supervisor that could be obtained from the collective research. A busy supervisor would also often conduct group meetings so that appointments could be done effectively within the same field of research.

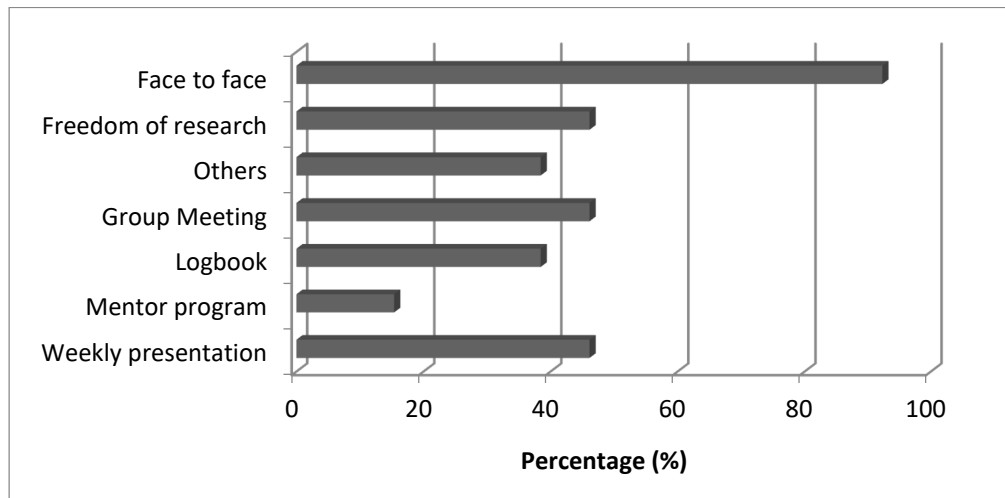

**Figure 8.** Percentage of monitoring and supervising methods implemented by lecturers.

Some lecturers prefer weekly presentations not only to know the progress of their students' research but also to develop their communication and organizing skills. Presenting on a weekly basis would also develop self-esteem, especially among international students who usually have difficulties linguistically. Logbooks and mentor programs scored low compared to other methods. Culture of excellence proved to be popular among lecturers based on Figure 9. Most top local universities have recently adapted culture of excellence for their faculties in order to increase their rank in the academic world. Some aspects that are included in a culture of excellence are excellent leadership, socialization, congruency development, and so on. The implementation of culture of excellence also proved to be beneficial as there was an increment in productivity when the technique was first introduced. This is explained in detail later in Figure 10. The publication stimulation technique scored the second highest with lecturers and followed by multi-dimension assessment and effective techniques, in that order. Multi-dimensional assessment is a new technique practiced by a few lecturers in a local university. A small group has been established consisting of lecturers with different expertise. The group emphasizes producing research with high-quality findings and results. Some of the techniques have been applied by the head of the group on students for two years and the performance in producing good quality students and publications has tremendously increased. One of the techniques in 'others' in Figure 9 mentioned by a number of lecturers is monthly presentation. This technique is suitable for lecturers who find it hard to schedule weekly presentations due to insufficient time.

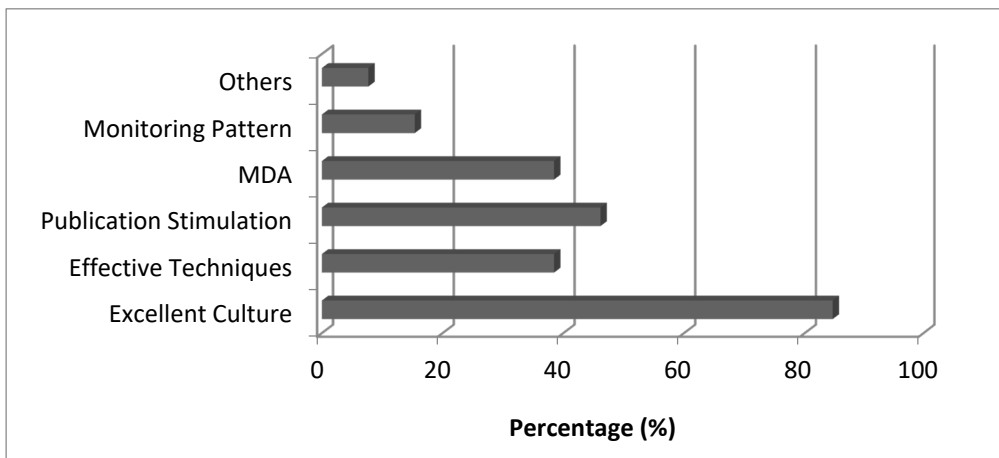

**Figure 9.** Percentage of research management techniques applied by lecturers.

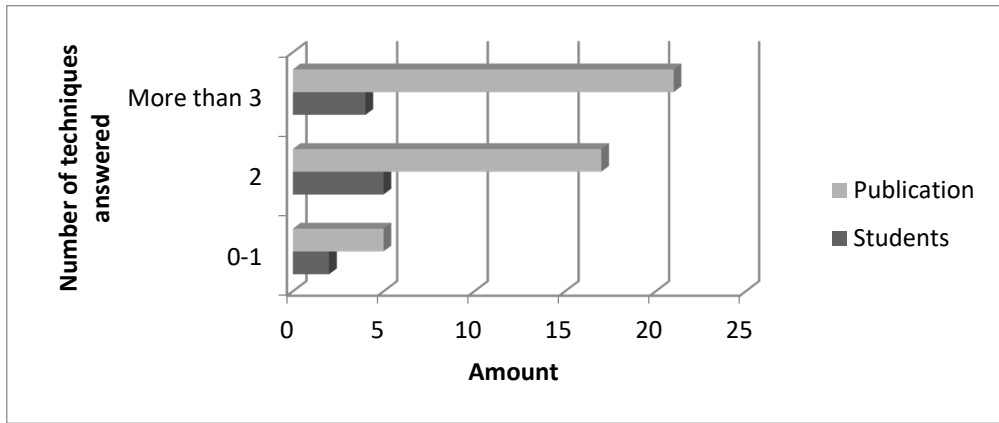

**Figure 10.** Number of students and publications produced by lecturers per year.

The collected questionnaires for lecturers were then categorized into three different groups based on the number of methods they have implemented in managing their research. This does not intend to put into question the lecturers' reputation and capability in research management but only to link the methods with the performance. Figure 10 shows that lecturers who answered one or fewer methods came up with the lowest number of students and publications. The number increased for lecturers who answered two methods and, for those who answered three and more, the number of publications produced per year were the highest. However, the number of students graduated per year was lower than for those who answered two methods. Lecturers who applied as many techniques and methods in their research management are more focused on publishing their research findings, which explains the huge difference in the number of publications. This is a good sign for lecturers if they really want to produce a high number of publications. Not only it is beneficial for the faculty, but it can also contribute to a prosperous and a more innovative community for a better future. The sustainability supervisory technique leads to effective monitoring, reduced workload, and increased productivity.

Based on the list of outputs in Table 1, some criteria scored low for researchers who implemented fewer than two techniques. For O1, only one researcher managed to produce at least one article in a first-quartile journal and several high-impact ones. Other researchers who managed to produce more than 21 publications and 5 students per year practiced more than three techniques featured in this paper. Most of the published articles by these researchers managed to be featured in this paper. Most of the published articles by these researchers are included in high-impact journals and the rest are at least in indexed ones. Researchers who answered two techniques and below produced less in terms of quantity and quality of publications (more papers published in indexed rather than impacted journals). However, the concern is on the level of quality the students pose in their research, and this depends on the satisfaction of researching. It contributes to the success of the research and eventually, the publication. Lecturers who did not use any technique scored the lowest in all aspects. This is mostly due to the fact that these lecturers were given more workload on teaching and management rather than research. Even though they had high commitment in these works, they still had minor research that needed to be done, especially at the undergraduate level. This also represents the same for O2, O3, and O4, where some lecturers focused only on delivering the number of publications generated regardless of the level, whether impacted or indexed. In this case, most were in the indexed article and proceedings papers categories which are lower than impacted articles. O4 and O5 involve the publication of academic-related books, mostly where the majority of the respondents were senior professors who are renowned researchers in their respective fields. Most of the books published were related to their own academic fields, research management, and motivation. Even though most of the books were a combination of more than one author, some were written by only one. O6, O7, and O8 put great emphasis on the generation of graduates and postgraduates. Even though the production of excellent

graduates does not rely directly on the lecturer's own performance and depends mostly on the students themselves, the lecturer has a crucial role in making this happen especially for postgraduate students [33]. It is at this stage that most students have decided which major they want to pursue, and lecturers are supposed to guide them on the correct path. It is common for most students to feel timid when communicating with their supervisors as compared to when they communicate with their peers. This has an obvious effect on O10, O11, O12, and O13, when the students' research output wins awards at various levels. These awards do reflect on the capability of the students as well as the effectiveness of the supervisor's method. A lot of efforts are put into the events held to showcase the finished products or research findings and combined with a decent presentation; these are the qualities that determine the performance level of the students. O14 is another important output to indicate a high-quality management profile. Even though the expenditure of the grant money reflects the research group as a whole, it is up to the head supervisor to control the money flow.

## 6. Conclusions

Although the primary concern for most of the students who raised the question of participation and interaction is the research criteria, it is nevertheless true that, like all human interaction, contact with lecturers and other students in the research laboratory is a social phenomenon in the postgraduate world. It is undeniable that some students are yearning simply for more social interaction with their fellow students and teachers. The students who emphasized the motivational aspect of feedback, and received good feedback from supervisor as encouragement, as well as those who pointed to feedback as a factor in reducing anxiety or a source of encouragement, were also concerned with the learning process. When students say that they feel encouraged by generous feedback, they appear to be referring to intellectual encouragement. It can be interpreted that the students are trying to say that this kind of interaction and engagement via close monitoring with the lecturer or supervisor enhances their confidence in their own academic ability, and therefore increases their enthusiasm and motivation for the research topic and perhaps for other intellectual work. This would lead to high quality research output. Moreover, their comments about the importance of feedback as an emotion regulator/reducer of stress seem for the most part to be concerned with their learning, where anxiety and other negative emotions are taken to be an obstacle to engagement with the discipline [34]. Therefore, comments or feedbacks from the supervisor are valuable for the student to move forward confidently and to guide them to maximize the outputs [32]. Simply being concerned is not enough but must be complemented by sustainable supervisory technique that physically and emotionally affect the performance of the supervisee or students.

Meanwhile, one possible situation where a busy lecturer might fail in the latter case, especially when managing a large group of students, is by treating students merely as names on a very long list [35]. This can be illustrated by the students' requesting comments and feedback on their research progress and the lecturer simply responding with a grade, with no comments or perhaps with only perfunctory ones, and with none that engage with the student's individual insights or mistakes. It is of course academically helpful to the student if feedback is individualized, but it is suggested that failing to provide personalized feedback is not only academically unhelpful but can also be experienced by the student as personally undignified. With a high number of postgraduate students and high student-to-staff ratios, students can easily feel isolated—they always feel like a nameless member of a featureless group [36]. However, it is important that students do not treat this as an excuse to fail in their research. Students' insistence on respect/recognition is a signal to lecturers that feedback is a way for them to address these feelings of isolation by engaging with students as individuals. This engagement will naturally revolve around the academic subject being assessed, but nonetheless, it can also serve the social function of engaging with the student in a conducive environment with a one-on-one relationship that the student values for its own sake. The supervisor is encouraged to explore an effective

monitoring system which could help them to engage with the student efficiently [37]. A model with a lot of monitoring can be adopted to monitor the research progress and keep monitoring the students' work. An sustainable supervision can also develop the most recommended attributes required by the stakeholders through effective activities and reliable assessment [38].

It seems that the quick, easy, and unethical approach to research by students frequently trumps a careful and thorough review of material. Easily accessible information downloaded freely over the Internet by using a search engine has become a popular trend among students in these modern days. There are also many publishers who offer free downloads or give open access to a full and complete article which can speed up the study of the literature. It could be an advantage to students to provide one comprehensive and holistic view about the subject matter. However, reviewing just the few pages or two of the results can usually expose adequate information to satisfy the minimum requirements for their papers and the information students collect is often questionable. This is further aggravated by their lack of critical evaluation of the material credibility they have secured. More frequent communication with faculty and students regarding available resources should be undertaken, as well as instructions to improve both effective and efficient access to information and discussions regarding criteria which can be used to evaluate data gathered for quality and suitability.

The study has provided some insights into students' research behaviors and perspectives and identified opportunities for supervisors to improve the research monitoring, as well as areas where additional strategy is needed to improve research skills among the students. There is a potential for follow-up research to this study by investigating faculty requirements for the type of research required in completing their projects, in order to balance the students' perspective gathered in this survey. By gathering a faculty perspective and input, a new path for collaborative efforts to improve the techniques in research management and research tools can be undertaken, as well as ensuring that grant funding includes all necessities to assist students in successfully conducting their research and becoming life-long researchers. Creating healthy competition can not only help us to reach our goals but can also make the process a fun one for students as well as improve their competencies. One way to do this, while also building teamwork and collegial collaboration, might include establishing a prize that everyone will receive or share if the institution meets a particular sustainability goal.

**Author Contributions:** Conceptualisation, A.H. and S.M.Z.; Supervision, M.S.A.-R.; project administration, R.A.-S. and A.Z.; Validation, A.K.A.M.I.; review and editing, I.H.K.; resources, M.J.M.N., K.O. and R.I. All authors have read and agreed to the published version of the manuscript.

**Funding:** This research was funded by Universiti Kebangsaan Malaysia through Research-University grant (KRA-2018-028).

**Institutional Review Board Statement:** Not applicable.

**Informed Consent Statement:** Not applicable.

**Data Availability Statement:** Not applicable.

**Acknowledgments:** This research was conducted in the Broadband, Network & Security Laboratory, Universiti Kebangsaan Malaysia (UKM).

**Conflicts of Interest:** The authors declared no potential conflict of interest with respect to the research, authorship, and/or publication of this article.

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
