# Peer review of "How Does Supervision Technique Affect Research? Towards Sustainable Performance: Publications and Students from Pure and Social Sciences"

_sustainability, doi:10.3390/su14095696_

Round 1

Reviewer 1 Report

Dear Authors,

Please find below some suggestions that could improve your manuscript.

Please make the research problem clearer in the Abstract. It is mentioned that relationship between techniques and performance. Please define performance. The aim should be stated at the beginning of the abstract.

Lines 48-51: Please add additional references in support of this: The number of publications does not measure the greatness of an academic. It is the quality of a particular article that is sought after. This phenomenon does not only cover a certain field, for example engineering, but also other fields ranging from psychology, history to chemistry and mathematics.

Lines 52-54: It is crucial for researchers to publish scientific articles because it is a key measure of their performance, and will also enable them to get research grants, promotion, professional networking and salary increment (Brinn & Jones, 2005). Please add more recent references in support of this. There is a vivid debate on the assessment of researchers (see the latest paper of LERU, white papers drafted by CESAER, etc.).

Line 205: Please explain more in-depth the ‘concept of publications and students of quality’. The paper needs to highlight better its focus: it is about student performance or researcher performance? If the paper relates to student performance, then it is needed to have a clearer approach to the concept.

In the same line of thoughts: `A publication that boasts quality is one that is packed with understandable and functional information, free of grammatical-error and has the potential to boast a significant amount  of citation’ – is this referring to student publications? What type of publication would it be?

` This specific publication also has to be reviewed by not only peers but also by other renowned experts.’ – I would suggest removing this affirmation.

Section 2. The Concept of Research Performance needs to be revised. Please make clearer the focus; researcher performance or student performance?. Then define it and analyze relevant literature on the topic. What is the theoretical framework underpinning the research?

Could you please add some references in support of these indicators? - Table 1 Outputs for High Quality Research Management Profile

Lines 221-223 - There are two aims from this paper; the first one is to evaluate the quality and quantity of publications produced by applying some defined techniques and the second one is the quality and quantity of students graduated. – Please revise and reformulate based on the above suggestions.

What do you mean by `sets of questionnaires? Have you applied two different questionnaires or more than two? More information about the quality metrics of those questionnaires is needed. Please present the structure, nature of items, validity, etc.

Additionally, more information on the selection of participants is needed.

Line 247 – Please explain what you mean by ‘samples’. Previously it was mentioned that you applied the questionnaire to 15 lecturers. Have you selected more than one sample?

The description of the techniques should be in the first (two) sections, not in the methodology section (lines 258 – 391).

The values in Figures 4, 5, 6 and 7 are percentages? Please mention the volume of the sample (n= …). The analysis needs to be revised and provide more in-depth information. The results need to be discussed in relation to other studies.

Please check for English grammar and vocabulary errors. For instance, research is not used at plural as `researches’.

Thank you for considering my suggestions!

Author Response

All comments have been amended successfully.

The word 'researches' can be acceptable and we ask your opinion either to change or maintain. This word can be found in many articles.

Reviewer 2 Report

Abstract should comprise : short background, specific problem to be handled, solution, method and finding. 

Introduction should include key publication as the concern of the work.

The authors should included questionnaire design, completed with hypothesis and research design for the survey. Reliability, validity and structural model is better to also be conducted and reported.

The important finding related with the hypothesis should be highlighted in both conclusion and abstract.

Author Response

The comments are addressed succesfully.

Round 2

Reviewer 2 Report

The reviewer's comments are simple yet essential, making sure about the clear novelty of the work. Authors provide no responses to initial comments of the reviewers. It is better to provide responses to each comment of the reviewers.

Author Response

 Respond to Reviewe

Comments

Amended

Lines 48-51: Please add additional references in support of this: The number of publications does not measure the greatness of an academic. It is the quality of a particular article that is sought after. This phenomenon does not only cover a certain field, for example engineering, but also other fields ranging from psychology, history to chemistry and mathematics.

The additional reference has been added.

(Brinn & Jones, 2005)(Ciencia & Coletiva, 2015).

Please make the research problem clearer in the Abstract. It is mentioned that relationship between techniques and performance. Please define performance. The aim should be stated at the beginning of the abstract.

Additional sentence has been added to clearer the problem statement.

Supervisory without effective monitoring and strategy planning can lead to zero output. The fear of productivity losses, mingling with the horror of massively declining performance, has encouraged many leaders to ramp up their sub-ordinates’ monitoring efforts

Lines 52-54: It is crucial for researchers to publish scientific articles because it is a key measure of their performance, and will also enable them to get research grants, promotion, professional networking and salary increment (Brinn & Jones, 2005). Please add more recent references in support of this.There is a vivid debate on the assessment of researchers (see the latest paper of LERU, white papers drafted by CESAER, etc.).

The additional reference has been added.

(Brinn & Jones, 2005)(Ciencia & Coletiva, 2015).

Line 205: Please explain more in-depth the ‘concept of publications and students of quality’. The paper needs to highlight better its focus: it is about student performance or researcher performance? If the paper relates to student performance, then it is needed to have a clearer approach to the concept.

Additional info added up

The performance of student progress can be measured via number and quality of publication. It is reflective to the content and knowledge, idea presentation, deliberation, critical thinking and etc.

In the same line of thoughts: `A publication that boasts quality is one that is packed with understandable and functional information, free of grammatical-error and has the potential to boast a significant amount of citation’ – is this referring to student publications? What type of publication would it be?

It has been referred to index and impact factor journal.

This specific publication also has to be reviewed by not only peers but also by other renowned experts.’ – I would suggest removing this affirmation.

Done reverted

New sentence:

Most of the high-quality publication are reviewed by field experts.

Section 2. The Concept of Research Performance needs to be revised. Please make clearer the focus; researcher performance or student performance? Then define it and analyze relevant literature on the topic. What is the theoretical framework underpinning the research?

Students who are referring here those registered post-graduate studies and research-oriented method.

Could you please add some references in support of these indicators? - Table 1 Outputs for High Quality Research Management Profile

The indicator is proposed by our side.

Lines 221-223 - There are two aims from this paper; the first one is to evaluate the quality and quantity of publications produced by applying some defined techniques and the second one is the quality and quantity of students graduated. – Please revise and reformulate based on the above suggestions.

We take note on this suggestion.

What do you mean by `sets of questionnaires? Have you applied two different questionnaires or more than two? More information about the quality metrics of those questionnaires is needed. Please present the structure, nature of items, validity, etc.

The two questionnaires differently to two group of the respondent (students and lecturers)

There are two aims from this paper; the first is to evaluate the quality and quantity of publications produced by applying some defined techniques and the second is the quality and quantity of students graduated.

Additionally, more information on the selection of participants is needed.

The selection randomly to pure and social science s student.

Line 247 – Please explain what you mean by ‘samples’. Previously it was mentioned that you applied the questionnaire to 15 lecturers. Have you selected more than one sample?

Samples referring to the respondent

The description of the techniques should be in the first (two) sections, not in the methodology section (lines 258 – 391).

The values in Figures 4, 5, 6 and 7 are percentages? Please mention the volume of the sample (n= …). The analysis needs to be revised and provide more in-depth information. The results need to be discussed in relation to other studies.

The samples did mention in the text. 15 Lecturers and 100 students.

Please check for English grammar and vocabulary errors. For instance, research is not used at plural as `researches’

The articles had submitted to editor.

Researches can also be used, however we change from researches to research

Abstract should comprise : short background, specific problem to be handled, solution, method and finding. 

Completely addressed

Introduction should include key publication as the concern of the work.

Done addressed

The authors should included questionnaire design, completed with hypothesis and research design for the survey. Reliability, validity and structural model is better to also be conducted and reported.

We choose no included all the info. It will extend the pages of the article.

The important finding related with the hypothesis should be highlighted in both conclusion and abstract.

It has been highlighted in the conclusion.

Round 3

Reviewer 2 Report

The article can be accepted for publication.

This manuscript is a resubmission of an earlier submission. The following is a list of the peer review reports and author responses from that submission.